# Innovation of Contemporary Chinese Urban Community Governance under the Perspective of Social Capital: Participation of Multiple Subjects Based on Community Proposals

**Dexin Wang and Shijun Li \***

School of Marxism, Chengdu University of Technology, Chengdu 610059, China
\*   Correspondence: shijunli1111@163.com

**Abstract:** To improve the modernization of social governance, the Chinese government has invested a large number of resources and policies into the field of community governance in recent years. This study takes the community proposal in China's Experimental Zone for Community Governance and Service Innovation as research cases, conducts a multi-case comparative analysis, applies social capital theory, summarizes four different community governance models from the differences of emotional and institutional social capital, and individual and collective social capital, and analyzes the process of community governance and proposal operation with the participation of multiple subjects. The study finds that community proposals expand the possibility of public participation, which is an extremely important reference value for the process of urban community governance and social democratization in contemporary China. However, community proposals are still policy-oriented, administrative, and benchmark-oriented, and the participatory roles and functions of multiple subjects remain unbalanced. The key to contemporary urban community governance in China lies in tapping community social capital, expanding the dimensions of social capital, and creating a sustainable mechanism for social capital transformation.

**Keywords:** diversified participation; urban community governance; social capital; community proposal

## 1. Introduction

At the end of the 20th century, the "community renewal" movement launched a campaign to implement communalism through bottom-up methods, and encourage community residents to participate in community governance, so as to restore community vitality and promote government reform and social development [1] (Putnam 2000). Under the influence of the movement, the community care development program, and neoliberalism, diverse community participation has become the new picture of community governance [2]. The government has transferred the right to provide public services and participate in community affairs to the community through the purchase of services [3]. With the emergence of community development corporations, volunteer organizations, social enterprises, and social organizations, communities have been "rediscovered" [4], and community participation has shifted from an emphasis on individual rights to a focus on the balance of social and collective responsibility [5].

In this process, community governance has been given a new connotation. Community governance is the process and behavior of supplying public goods, managing community affairs, and meeting community needs through consultation and negotiation, coordination and interaction, and collaborative action based on certain consensus and public interest, with the community as the territorial scope and stakeholders in the community as the main body, i.e., diverse subjects participate in the daily management of the community, with the common goal of improving community life [6,7].

In recent years, the research on the relationship between the state and society has gradually extended to the discussion of the role and status of multiple participating subjects, the coordination and cooperation among participating subjects, and how to enhance the effectiveness of the participation of different subjects. First, relevant research focuses on the participants and their functions. The participation of multiple subjects in community governance is the trend of social governance development (Community governance is the microscopic area of social governance, and community governance is the focus of social governance, the nerve at the end of governance), and which functions different subjects take on, how to maximize their respective effectiveness, and how to collaborate are the primary issues of public participation system design [7,8]. Second, part of the research focuses on analyzing the factors that affect participation behavior. At the micro level, there is a direct influence of individual's knowledge endowment, cognitive perception, and economic ability on the impact of public participation [9]; at the macro level, regional differences, social environment, and political system have a certain guiding role. Third, the qualitative and quantitative evaluation of the effect of public participation is also the focus of the theoretical and practical circles. Public participation has significant effects on conventional governance in micro areas such as environmental transformation, space creation, public service provision for the benefit of the people, and trust and identity construction, and can play a positive role in the face of complex governance issues such as pollution management and public safety; however, there are still many limitations [10]. For example, the public has limitations in understanding scientific issues, individuals are easily driven by interests to take irrational actions, and the method of public participation is relatively simple or formalized. Fourth is the study of ways to enhance the capacity of public participation. To enhance the effectiveness of public participation in community governance, it needs to be enhanced at the levels of political ecology, institutional innovation, democratic consultation, citizen science, mass mobilization, information technology, mechanism guidance, social incentives, and fair participation [11–14].

As a modern country with a huge population, China's grassroots community governance faces unique problems, such as a large population base, a large floating population, and tensions between supply and demand of public resources [15–17], which, to a certain extent, make the traditional state–society theory and public participation theory insufficient explanations. To further explain the relationship and coordination between the state and society in community governance, this study followed an experimental process of community governance and service innovation in China, the theme of which was to explore the mechanism of community proposals. The researcher experienced the whole process of community proposals as a reviewer of community proposals. The model of community proposals has great research value, which is useful for exploring the hindrances of public participation in urban community governance in China today, on the one hand, and for providing insights into innovative community governance approaches in mega-populated cities on the other.

## 2. Research Construction and Case Selection

### 2.1. Research Construction

Both in the development of community governance theory and the evolution of governance practice, the relationship and the distribution of power between the state and society are central issues [18,19]. The theory of metagovernance emerged from the reflection of government failure and market failure. Metagovernance aims to resolve the conflicts between government, business, and civil society in the process of joint participation in order to promote synergy and complementarity among multiple governance actors, i.e., to perfectly combine the models of bureaucratic governance, market governance, and network governance through institutional design and management mechanisms [20]. The purpose is to clarify the positioning of governance subjects such as government, market, and civil society, to combine them effectively, to fully integrate the resources and strengths of each governance subject, and to construct macro-arrangements for the organization, system, or

mechanism of each governance model to form a new governance mechanism [21]. At the same time, to avoid the failure of collective governance, the key role of government needs to be clarified [22]. This is also seen by opponents as a return to authoritarian government models [23].

Government dominance can pool resources, integrate the public interest of stakeholders, and maintain social equity; however, this model of community governance has resulted in a loss of flexibility and autonomy in community life [24], as well as some bias in decision making and moral hazard [25,26]. Cooperative governance theory, on the other hand, suggests that the cooperation of multiple actors can create a "structure–response" mechanism to collect a wider range of public opinion and achieve an effective response from the government to the public [27–29], which is an important way to empower the public [30].

In terms of community governance structure, community governance contains multiple actors, such as the state, citizens, social organizations, and the market [4]. The process control of community governance is to adjust the community governance structure according to the real conditions to achieve a better governance result. Therefore, community governance with diverse participation is essentially concerned with two points: first, which roles actors such as government, citizens, social organizations, and informal organizations or institutions play in community governance, and how to maximize the functions of different actors [31,32]; second, how to coordinate and cooperate with multiple actors in the process of community governance, so as to produce good governance effects [33].

However, in the practice of community governance, the phenomenon of community failure also occurs frequently. Diversified community governance faces the dilemma of power distribution and use. On the one hand, the government's empowerment of society is subject to both internal resistance and pressure to maintain social problems. On the other hand, the community appeals for more power transfer, but often lacks the corresponding capacity and resources to assume the responsibility of autonomy [34,35]. Therefore, theories of metagovernance and cooperative governance need to be revisited during community governance practice.

Due to historical reasons, spatial scale, age structure, population mobility, cultural habits, social resources, and other factors, the level of participation of Chinese residents in community governance varies, and the effectiveness and status of multiple actors in the process of community governance are also very different, and these differences are essentially reflected in the level of social capital endowment possessed by the community itself [36,37]. Therefore, the characteristics of community social capital influence the "structure–process" analytical framework of community governance [38].

Social capital theory focuses on the sum of all the resources that social organizations can utilize to achieve their goals in a network of social relationships. According to Putnam, "social capital refers to the characteristics of social organizations, such as trust, norms, and network, which can enhance social efficiency by promoting cooperative behavior." [1] Based on the local characteristics of community governance in China, social capital theory answers the dilemma of "weak participation" in the public sphere in terms of the participation dynamics of community governance subjects, the participation mechanisms constructed within the community area, and the interaction of the community's multi-participation relationship network.

The process of establishing the social capital of a community includes three steps: the first step is to build trusting relationships. Based on informal neighborhood interactions, trust relationships are established between neighbors, community residents, and external actors to catalyze public understanding and awareness of community governance [39]. The second step is to formulate procedural community norms. In the context of common life and common interests, effective norms and rules are developed from the bottom up through continuous actions to maintain order within the community. The third step is to build a broad social network. The focus of social capital is on the overall development of the community, and the key is to obtain the social network based on the relationship

resources. The more freely the resource can be converted into collective action, the stronger the social capital.

From the perspective of the structure of social capital, this paper divides the attribute tendency of social capital into "individual social capital" and "collective social capital" based on "self- or society-centered", and "emotional social capital", and "institutional social capital" based on "soft or hard capital accumulation", and constructs a new "structure process" analysis framework based on it (Figure 1).

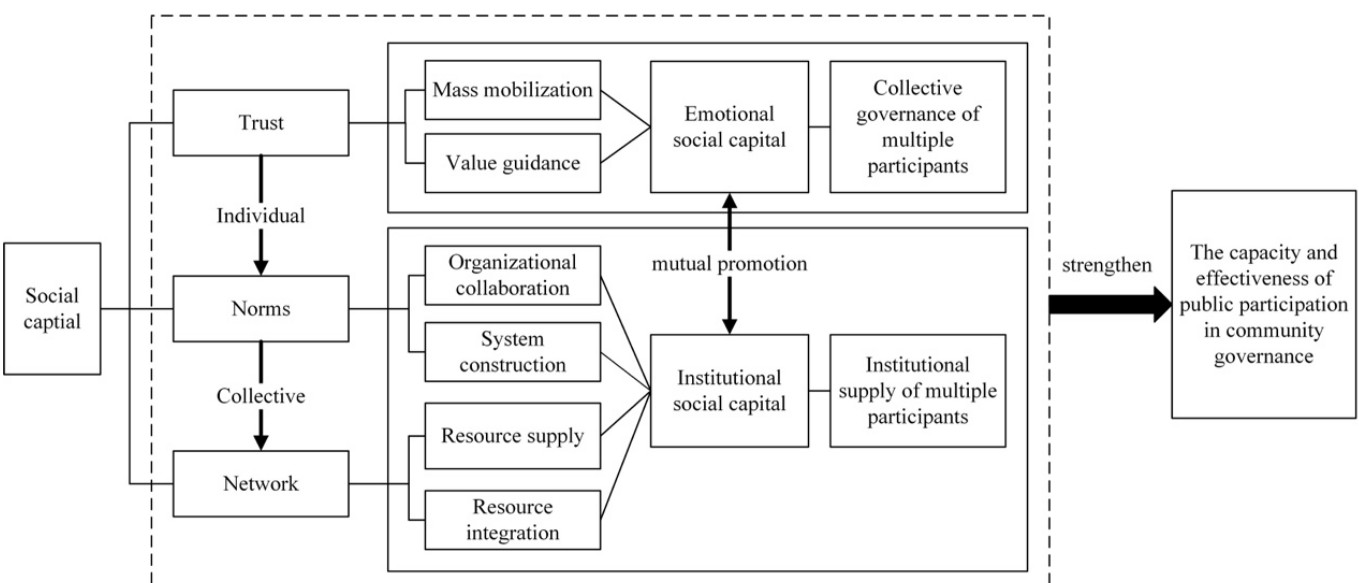

**Figure 1.** "Structure–process" analysis framework of community governance based on social capital.

Individual social capital refers to resources such as power, wealth, prestige, information, opportunities, and knowledge that originate from, and are embedded in, an individual's external social network and facilitate individual actions. Collective social capital is an institutionalized and organized association that transcends the individual, and is a collection of value resources of all members of the group. Emotional social capital includes both the individual's own emotional capabilities, such as feelings, beliefs, will, and values, and other implicit resources, as well as the emotional support, reciprocal relationship, and sense of community formed in the process of interaction. Institutional social capital refers to the collection of all rules rationalized or legitimized in the process of public participation in community governance, and the providers of community rules can be either authorities or community residents.

The stock of social capital a community possesses influences the governance functions that pluralistic participating subjects need to supply to the community, which determines the basic governance structure of the community. In China, diversified participating subjects, such as the government, community committees, Communist Party organizations, residents, enterprises, social organizations, and informal organizations in the community, assume the functions of resource supply, resource integration, organizational collaboration, system construction, mass mobilization, and value guidance [40]. The cooperation of each participating subject creates the basic framework and action guidelines for institutional supply and collective governance. In the process of participating in community governance, with changes in participating subjects, participating resources, and participating goals, a sustainable and diversified participation structure will tend to accumulate more social capital and create and maintain a dynamic balance of more diversified social capital. Institutional social capital and emotional social capital promote each other, and this mutually integrated and mutually supportive transformation process strengthens the capacity and

effectiveness of public participation in community governance, which in turn shapes the structure of grassroots community governance in China.

*2.2. Case Selection*

As one of the mega cities in China, Chengdu, with a population of 21 million, is a typical city with a large community system (China's community governance is a process that takes the community as the regional boundary, and takes community committees, community residents, social organizations, social workers, enterprises and commercial establishments, government, and other stakeholders as the main body. It aims to mobilize the public to participate in the daily management of the community, carry out voluntary services, cultural activities, etc., to improve community life. China's urban communities are characterized by a large geographical space, a large number of permanent residents, a high proportion of residents in the floating population, and a high degree of overlap between business space and residential space; thus, they are called large community systems. China's local administrative system is divided into four levels: province, city, county, and township. The Street ("Jiedao" in Chinese) is the township-level government management organization. There is no administrative level in the community, but community workers are generally recruited by the Street. Community committees carry out urban management and service work under the management and leadership of the Street) [41]. According to the Master Plan for Urban and Rural Community Development and Governance in Chengdu (2018–2035), the reasonable number of urban and rural communities in Chengdu in the future ranges from 3900 to 4200, and the spatial scale of communities is controlled at 15,000 people per community. As one of the 31 national community governance and service innovation pilot zones, Jinniu District of Chengdu City has put forward the community governance mechanism of community proposal. The community proposal (Such community proposals are not political consultation proposals put forward by party representatives, people's representatives, or members of the Chinese People's Political Consultative Conference. Community proposals are not political consultations, but community-wide public interest consultations) carried out in Jinniu District is open to all communities in the district starting from April 2021. The community proposal process consists of five steps. The first step is to form the idea of community proposal. The stakeholders of community-level public affairs form the preliminary idea of the proposal around community-level public issues. The second step is to put forward the formal proposal. This is achieved by filling in community proposals online and offline to form written community proposals. The third step is to establish the community proposal. In accordance with the prescribed procedures, the community proposal committee (organized by the community committee (Community committee is called Jumin weiyuanhui in China. According to the provisions of the Organic Law of the Community Committee, its team consists of five to nine members, including the director, deputy director, and members. It is elected for a five-year term, and its members can be re-elected. The community committee can be divided into several resident groups, and the group leaders are elected by the resident groups. The working funds of the committee shall be prescribed and allocated by the governments at higher levels. The main tasks of the community committee include (1) to promote The Constitution, laws, regulations and state policies; (2) to safeguard the lawful rights and interests of residents; (3) to educate residents to fulfill their obligations under the law; (4) to care for public property; (5) to carry out various forms of socialist spiritual civilization construction activities; (6) to assist in the handling of public affairs and public welfare of residents; (7) to mediate civil disputes; (8) to assist in safety publicity; (9) to assist the government or its organizations to conduct work related to the interests of the residents; (10) to reflect the opinions, requirements, and suggestions of the residents to the government or its organizations; (11) to carry out community service activities for the benefit of the people; and (12) to establish related service businesses; and so on) shall hold a meeting to discuss whether to file a case, and put forward problems and related objectives. The fourth step is to solve the community proposal issues. After the case is filed, the community proposal committee will notify

the relevant stakeholders to jointly negotiate and resolve the proposal. If the lower-level proposal organization cannot solve the problem, it shall be submitted to the higher level committee for deliberation. The fifth step is to collect the community proposal feedback. After handling the issues, each participant shall fill in the feedback form.

As of October 2021, the community proposal platform received a total of 190 proposals online, including 45 in construction, 32 in remediation, 6 in public safety, 44 in community services, 56 in self-governance, and 7 others. Subsequently, the government of Jinniu District reviewed 80 community proposals that had been properly resolved. On-site reviews are conducted by representatives from relevant government departments, community residents, and social organizations; online reviews are conducted by experts and scholars in the fields of community governance and consultation and deliberation from across the country. The online and offline judging panels generated a comprehensive evaluation of the declared cases from the aspects of community proposal, residents' participation in consultation, action implementation, and result effectiveness, and provided relevant opinions and suggestions. The 30 outstanding cases were finally selected (see Appendix A for the scoring rules for community proposals).

The authors conducted the review as experts in the evaluation of community proposals. After the review, the authors visited the communities where the above award-winning cases were located from August to December 2021, and invited relevant proposal proponents, community staff, participating residents, representatives of social organizations and enterprises, community volunteers, and other informed individuals to conduct in-depth interviews, obtaining a large number of first-hand interview recordings, policy texts, and other relevant information. The main questions of the interviews include (1) who leads the community proposals, which groups are the main participants, and what role each plays; (2) how to obtain the resources needed in the community proposals (funding, manpower, policy support, etc.); (3) what were the main deliberative organizations in the community and what was the deliberative process; (4) what difficulties are encountered in the process of promoting the community proposals and how to solve them; (5) how effective is the community proposal; (6) what are the proposals for further improvement; etc. The data of this study come from two sources: the first comprises about 200,000 words of recorded manuscripts obtained through interviews; the second is the text materials of 30 award-winning excellent cases collected on the community proposal platform, totaling 186,000 words.

For the research data, Nvivo software was used for data processing. The recording data were deeply mined through thematic analysis, and the normalized case text materials were subjected to phrase frequency counts through content analysis. This formed a database of community proposals. Through coding and organizing these data, the study found that different community social capital characteristics lead to different community governance approaches. Based on the differences in the endowment of community social capital and the characteristics of the dominant participants of community proposals (Figure 2), this paper selects four types of community proposal models (the four types of community proposal models can basically cover all community proposals, including the seven proposals of Xiangxian governance, eight proposals of third-party governance, ten proposals of cooperative governance, and five proposals of a contractual relationship. Meanwhile, the proposals on environmental improvement (Community A), volunteer team building (Community B), community charity (Community C), and public space creation (Community D) belong to the basic issues of construction, community services, self-governance, and remediation, respectively, and these proposals account for 93% of the total proposals. Using 30 excellent cases, our study compares and analyzes the dominant participants, organizational basis, institutional basis, consensus basis, social capital tendency, public participation subjects, and the key functions of participants of the proposal.

**Group social capital**

Community B

A companion model under the
cultivation of social organizations

Third-party governance

(Volunteer team for people with
disabilities)

Community C
The commercial and residential
cooperative construction model under
the leadership of the community
committee

Cooperative governance

(Charity trading activities)

**Emotional social capital**

**Institutional social capital**

Community A

The spontaneous autonomy model of
the residents

Xiangxian governance

(Environmental improvement)

Community D

The joint model of government,
enterprise, and society led by
enterprises

Contractual relationship

(Public space creation)

**Individual social capital**

**Figure 2.** Classification of community governance models based on social capital tendencies.

## 3. Analysis of the Community Governance Model with the Participation of Multiple Subjects in Contemporary China

*3.1. Xiangxian Governance ("Xiangxian", also Known as Virtuous Villagers, Refers to the Local Community Residents Who Are Virtuous, Talented, Prestigious, and Deeply Respected by the Local People in China): The Spontaneous Autonomy Model of the Residents*

Community A is a commercial housing community consisting of 12 low-rise commercial buildings, opened in 2001, including 10 residential buildings, 22 units, more than 1200 residents, and 80 surface parking spaces (no underground parking). Between 2006 and 2014, the community had experienced 8 years of "no management", a history of chaos, and the living environment and human environment was relatively poor. In 2014, a group of enthusiastic residents set up a courtyard self-governance committee, and put forward a proposal to beautify the neighborhood environment, including repossession of privately occupied houses, renovation of neighborhood gates, parking space renovation, beautification of the neighborhood environment, establishment of neighborhood homes, establishment of neighborhood self-organization, and revitalization of neighborhood homes. This series of initiatives has revitalized the social resources of the neighborhood, and realized the goal of common construction and governance in the community.

The resident-driven spontaneous self-governance mechanism is suitable for neighborhoods with more prominent individual social capital, relatively small and closed community space, and residents who have lived in the neighborhood for a long time; such neighborhoods often have potential emotional social capital, and it is easy to establish a sense of common feeling and identity among residents. Similar to most old neighborhoods in China, Community A suffers from aging and unreasonable design of public facilities,

underdeveloped public services, poor community environment, difficulty in attracting external resources, and weak ability to obtain social capital on its own. When the chaotic property management and dirty environment seriously affect residents' lives, the common governance problems faced rally the residents of the community to take the initiative to explore solutions to the problems. This spontaneous, collective action to defend their interests constitutes defensive participation in community governance.

In the early stage of community deliberation mechanism, the phenomenon of community members working separately was more serious. The initial intention of some residents was to solve the painful problems of community management, but due to the constraints of personal interests, ability, and information mastery, the community eventually introduced the lowest-priced property management company, which failed to take effective measures, and thus, community management fell into a more chaotic situation. In the face of the painful community problems that need to be solved, the president of the community yard committee, with his years of experience and ability in business management, his dedication to caring for his neighbors, and his unique personal charm, won the trust of the community residents, and took the lead in promoting the establishment of a more independent value code for the community. At the same time, stable informal organizations based on hobbies and interests have emerged in the community. The Grassroots Drama Club has built emotional networks among their members by performing together, holding dam parties (A form of folk gathering in Sichuan, China. The venue is extremely simple, usually with only benches and a few tables, and the participants exchange ideas in their native language in the form of a family conversation), and participating in volunteer activities, gradually expanding residents' sense of belonging and identity within the community.

However, there are still obvious drawbacks and limitations to such governance paths. First, they rely too much on individual leadership. Members of the community yard committee in older communities are generally retired and older, making it difficult to acquire new emotional capital. Second, the community committee has a single function, and fails to reflect a strong capital coordination capacity. Third, the sustainability of governance capacity is more fragile, and once the support of emotional capital is lost, residents' willingness to participate decreases significantly. Fourth, the community governance problems that can be solved are more limited. Although residents' spontaneous self-governance mechanisms are useful for routine governance in micro areas such as garbage sorting, environmental beautification, parking space renovation, and community space planning, it is difficult to control more complex, unexpected, and global issues, and cannot form a long-term endogenous community governance force.

*3.2. Third-Party Governance: A Companion Model under the Cultivation of Social Organizations*

Community B is an old community with 201 people with disabilities, including 80 people with physical disabilities, 40 people with mental disabilities, 34 people with visual disabilities, 23 people with hearing disabilities, 21 people with intellectual disabilities, and 3 people with speech disabilities, which is a relatively high number of people with disabilities. The social organization for helping the disabled in the community put forward a proposal to establish a mutual aid organization and an employment platform. A self-organized Sunshine Volunteer Service Team was established in the community, which is composed of people with disabilities and their families. By building a care service platform, a mutual help service platform, and an employment and entrepreneurship platform, the team provides targeted employment assistance, offers skill training, including Shu embroidery and Sichuan brush making, and organizes handicraft charity sales to re-establish the connection between disabled groups and the community.

In the process of promoting the community proposal in Community B, the key facilitator was the social organization for the disabled. It was not until the voluntary service team for people with disabilities established a stable operating mechanism that the organization gradually receded into the role of an observer. In this process, the community committee was more in the position of a resource linker. By highly empowering social organizations,

the professional capacity of social organizations for the disabled to participate in community governance has been released, effectively providing a convenient channel for certain groups to integrate into the general community environment and rebuild social capital.

The most important step for social organizations in the process of entering the community is to gain the recognition and trust of the residents. The social organization for people with disabilities in Community B has been able to successfully establish a volunteer team for the residents due to two aspects: At the material level, with the help of street and community party organizations, social organizations have built a platform to help people with disabilities start their own businesses, and community committees have provided places for people with disabilities in the community to learn and train, linking employment resources and providing legal protection, so that the target group has the space to develop their strengths; at the spiritual level, volunteer teams for people with disabilities are the main vehicle for public participation in the community. Through sharing experiences, teaching skills, and helping others, they inspire the emotional resonance of other people with disabilities, encourage them to integrate into the daily activities of the community, reconstruct social capital, and rely on the power of the group to influence those around them. The volunteer team combines the advantages of trust capital of families and associations, and through the understanding and care among families with disabilities, they transmit positive emotions to other groups with disabilities, other residents in the community, and other groups outside the community, thus forming a vibrant collective and emotional social capital.

The companion model under the cultivation of social organizations focuses on the collective excavation, training, and development of residents' self-governing organizations to assist residents in developing a sense of self-organization, the ability to grow independently, and the behavior of continuous self-governance. In general, community-constructed trust-based social capital is mainly derived from families and associations [42]. The former has a strong emotional component, but is somewhat exclusionary, while the latter tends to be a "voluntary affiliation" with more diverse and inclusive attributes, but also faces the risk of becoming a fragmented organization with weak institutional binding. Studies have shown that a good nurturing mechanism for social organizations has a significant effect on increasing residents' social capital. For communities with obvious common characteristics, professional social organizations that have operated well for many years can effectively create collective social capital.

However, the applicability of the companion model based on shared traits is limited. First, companion organizations formed by common traits tend to have strong internal cohesion, but are significantly less effective in attracting heterogeneous groups, making it difficult to establish a sustainable mechanism for introducing social capital. In particular, due to the unique personnel structure of Community B, it is difficult to replicate the model of revitalizing community capital using the volunteer service team for people with disabilities as an opportunity. Second, the generation and development of the companion model depends on the professional capacity of social organizations, and the independence, autonomy, and creativity of community residents themselves are weak. Social organizations need to continuously empower the public through emotional investment and construction. An excellent resident companion organization requires not only the coordination of external forces such as government, party organizations, and social organizations, but also the evolution of an institution and culture with its own characteristics, which is often a long and tortuous process of practice. Finally, the companion model nurtured by social organizations is also only applicable to daily and routine community governance issues, and its role is often limited to community-level public affairs related to its own group.

### 3.3. Cooperative Governance: The Commercial and Residential Cooperative Construction Model under the Leadership of the Community Committee

Community C is located at the North Railway Station, the starting point of the Chengdu-Chongqing twin city economic circle, with an area of 1.25 square kilometers,

relying on the geographical advantages of "one hub and two economic circles". The community is full of stores, and has a complex staff structure. There are 20,138 industrial units (including commercial stores or government and social organizations), including 33 professional markets and 15 government offices in the district; the resident population size is 35,884, of which the employees of the above units account for 70% of the resident population.

Community C has strong resource attraction due to its unique geographic, commercial, and population density advantages. Under the impact of the new digital economy and the continuous influence of the new crown epidemic, the pressure on the operation of physical stores has increased. To solve the problem of commercial development in the community, under the principle of "encouraging innovation and being tolerant and prudent", Community C held a forum with commercial establishments in the area to explore new ways to stimulate consumer creativity and serve community residents. Through demand research, market planning and activity connection, the community committee took "Lotus Charity" as the spiritual core to link up the common action between commercial establishments and residents. On the one hand, the "Lotus Charity" comes from local humanities anecdotes, which is a continuation of the community residents' spirit of protecting children and sponsoring education during the war period, and increases the sense of honor and identity among the community residents. On the other hand, activities such as charity collections, microfunds, microprojects, charitable activities, and cultural and creative designs held under the name of "Lotus Charity" form community brands and coordinate community business power. The institutionalized creation of consumption scenes is conducive to the establishment of community brands and the formation of a reciprocal value creation model between communities and business districts.

Through the advocacy of the community committee, Community C responded to the appeal of many commercial establishments in the community, led by the community committee, introduced the resources of social organizations, created a communication platform between commercial establishments, mobilized residents to participate, and encouraged social capital to feed the community charity projects by cultivating "righteous business", integrating the resources of stores, residents, and social organizations. In the process of organizing the fair, it creates a multifaceted scene and community brand that integrates community service, public welfare culture, modern industry, and living space. In the process of organizing charity bazaars, the community committee establishes a systematic process for transforming social capital. The charity bazaar attracts the participation of commercial establishments and neighboring residents with the concept of public welfare, and each charity bazaar condenses a theme. The community and social organizations select qualified commercial establishments or enterprises to enter the bazaar and recruit volunteers to maintain the order of the bazaar. A portion of the profits made by the commercial establishments through the charity sales will go to the community charity fund, which is used to help the disadvantaged and the needy, such as providing services for the elderly, environmental improvement, skills training, and so on. In particular, through childcare and education activities, the spirit of "Lotus Charity" is passed on to the next generation, and these beneficiary children become volunteers for subsequent charity activities one after another, so that the community brand of "Lotus Charity" can be continued and developed.

In the face of the large number and complexity of the participating parties, the work of the community committee has achieved good results, especially in building institutionalized business operation processes and charity operation regulations based on the community's own social capital, and creating emotional social capital by condensing the community brand to realize the diversified extension of the social capital dimension. However, the following conditions are required for the construction of this model: first, the organizational capacity of the community committee staff is high, and they need to possess rich working experience and the spirit of innovation; second, the community itself needs to have potential and untapped social capital [43], including public space for transformation, vibrant consumption capacity, geographic location for gathering popularity, etc.; third,

there is a need to explore the cultural value elements that are suitable for the human characteristics of the community, and that cover a wide range of common feelings, so as to realize the re-engineering of the community governance process.

*3.4. Contractual Relationship: The Joint Model of Government, Enterprise, and Society Led by Enterprises*

Community D covers an area of 0.7 square kilometers, with 33 compounds under its jurisdiction, 7586 residential households, and a population of about 24,000. Among them, there are 1130 senior citizens over 80 years old, 307 disabled people, 36 families with specialist support (57 people), 48 low-income households (59 people), and 1669 veterans. There are many scientific research units and troops in the area. There is also a commercial street running through the community. Every day, a constant stream of people come to the community committee to handle various affairs. However, the community office building was built in the early 1990s, with steep office stairs and old indoor hardware facilities, causing serious safety hazards.

Harbin Bank's Jinsha Branch, located in the community, took the initiative to submit a proposal to contribute the bank's public space to be shared with the community to provide services to residents. After the community committee proposed and collected residents' opinions, and the approval of higher government departments, it finally agreed to build a community "Dingzhi Space" (Dingzhi Space refers to participants in the community working together to create a community space) with Harbin Bank to carry out comprehensive public services. At the same time, the "Dingzhi Space" also has a "Happy Kitchen", a dance room, a children's area, a recreation area, a tea room, a recreation room, a meeting room, and other corners where residents can carry out a variety of activities.

In the process of creating the "Dingzhi Space", Harbin Bank is in the leading position. As a profit-oriented enterprise, Harbin Bank hopes to break the shackles of the traditional business model and achieve a win–win situation between economic and social interests, and took the initiative to cooperate with the community by funding and designing the space renovation plan. Through cooperation, Harbin Bank not only promotes the extension of the business environment to the community, but also the combination of commercial services and community welfare, to achieve the "resource sharing, complementary advantages, win–win cooperation" pattern of joint construction, opening up a new model of cooperation between government, enterprises, and the community.

To transform commercial behavior into a win–win model of social value creation, the community committee of Community D, together with the enterprises and commercial establishments in the area, has established an institutionalized space creation program. First, the institutional design is used to promote governance practices. Through the signing of party building agreements (Party building is led by the community party organization, where the community and the institutions, enterprises, and commercial establishments in the area are linked on an equal basis with common needs, common interests, and common goals to establish mutual ties between the party organizations of each unit. Through the communication of information and exchange of experience, all parties are coordinated to participate in community work), enterprises are introduced to the area and organize activities that meet the needs of community residents. The standardized convenience service activities and the bank's daily business activities are reasonably laid out, building a new scene of joint community governance. Secondly, the community committee is used to activate the multi-participation force. The key to the joint participation of multiple forces is trust. On the one hand, it can break the governance barriers arising from the fragmentation between society and enterprises, and integrate the fragmented social resources; on the other hand, the space overlap between the community committee service space and the bank's office hall can deepen the residents' trust in the enterprises. Finally, it promotes linking and creating social capital with livelihood services. The space of Community D is created based on the public's livelihood issues of most concern, which increases the interaction between

enterprises and residents, deepens the residents' impression of enterprises, and provides the possibility for enterprises to attract revenue.

The joint platform of government, enterprise, and community led by enterprises introduces the business operation model into community governance, and provides a new way of thinking for enterprises to actively participate in the governance of public affairs in the community. Community D matches the resources of enterprises in the community with the needs of community residents, using the public space of Harbin Bank as a venue to link other enterprise resources such as catering, training, medical care, education, and livelihood, providing residents with diversified public services while promoting the transformation of enterprise business models. This model breaks the one-way demand model of the community persuading enterprises to provide resources, and turns it into an interactive response model of enterprises making demands and the community linking resources, which not only gives enterprises more autonomy to choose, but also reduces the community's own governance pressure and effectively arouses enterprises' sense of community participation. At the same time, this is also a new attempt to respond to the transformation of new industries and promote enterprises to extend their business environment to the community.

The success of Community D's governance innovation is attributed to two aspects: Firstly, enterprises take the initiative to put forward their governance demands to the community. In the face of the impact of the "Internet +" economy, the traditional bank business model finds it difficult to resist the systemic risk; thus, Community D urgently needs to find a new marketing model. Harbin Bank's cooperation with the community is an innovation in community governance and an exploration of the enterprise's response to consumer transformation. Secondly, the model integrates business logic into the logic of governance. In the process of institutionalizing space creation, Community D adopted the enterprise's space design and activity operation plan, authorized professional activity planning to enterprises and social organizations, and ensured the sharing and public nature of activities with party construction, maximizing the effectiveness of multi-party participation. Therefore, this model is based on the premise that enterprises in the community are relatively strong and have idle resources. In addition to conventional governance issues, this model can also be applied to certain fine-grained public service provision issues; however, because the overly commercial model finds it difficult to generate emotional social capital, it may still fall into a governance impasse in the face of sudden crisis events (Table 1).

**Table 1.** Comparative analysis of four community governance models with the participation of diverse subjects.

| Model / Dimensions | The Spontaneous Autonomy Mode of the Residents | A Companion Model under the Cultivation of Social Organizations | The Commercial and Residential Cooperative Construction Mode under the Leadership of the Community Committee | The Joint Model of Government, Enterprise, and Society Led by Enterprises |
|---|---|---|---|---|
| Dominant participants | elite community residents | social organizations | community committees | enterprises |
| Organizational basis | loosely organized | more closed organizations | well organized | cooperative organization |
| Institutional basis | centralized decision making with the deliberative group as the core | collaborative participation, lack of participation in decision making | community-led, collaborative, and consultative | community-led and contractual |
| Consensus basis | acquaintance society | empathy | traditional Culture, business win–win | business win–win |
| Social capital tendency | Individual, emotional | collective, emotional | collective, institutional | Individual, institutional |
| Public participation subjects | unitary | unitary | diversified | single, with potential for diversification |
| Key functions of participants | mass mobilization, value guidance | value guidance, resource supply | resource integration, organizational synergy, resource supply, system construction, mass mobilization | resource integration, organizational synergy, system building |

**Table 1.** *Cont.*

| Model / Dimensions | The Spontaneous Autonomy Mode of the Residents | A Companion Model under the Cultivation of Social Organizations | The Commercial and Residential Cooperative Construction Mode under the Leadership of the Community Committee | The Joint Model of Government, Enterprise, and Society Led by Enterprises |
|---|---|---|---|---|
| Applicability | routinization of governance in the micro area | public affairs issues related to certain groups of people | routinization of governance in the micro area; complex, integrated community governance | routinization of governance in the micro area; integrated community governance |
| Limitations | single function, weak sustainability, vulnerable to personal profit maximization | lack of independence in the growth of organizational members; tendency to focus on collective interests rather than the overall interests of the community | requires strong working ability of community committee staff; richer potential social capital in the community | relatively strong enterprises and idle resources in the community; lack of mechanisms for generating emotional social capital |

(Source: compiled by the authors).

## 4. The Functions and Limitations of Multiple Subjects in Contemporary Chinese Urban Community Governance

### 4.1. Community Residents

Community residents are an important and integral part of public participation in community governance. However, the effectiveness of China's community residents' self-governance model is far below expectations. This is reflected in (1) low-quality participation behaviors, including lack of targeting, low level of participation, and monotonous forms of participation. Public participation in public affairs is rarely related to major decisions, except for some environmental decisions [44], and most of the deliberations are limited to community environmental improvement, community space creation, and public service enhancement projects involving food, clothing, housing, transportation, culture, sports, and fitness [14]. Meanwhile, when faced with refined or complex governance issues, the effectiveness and quality of residents' participation are often lower than those of NGOs, enterprises, and other organizations. (2) Subordinate participation behavior, for example, residents' participation is less active and less autonomous, and often driven by external forces such as communities, streets, party organizations, and community organizations. As a result, residents' participation is often passive.

In essence, due to the inefficiency of public participation, even though residents are geographically and spatially "present", they are still "absent" from decision making in community affairs. The reasons for this are as follows.

First, emotional connection is relatively weak. In the process of community-based reform, the social system of acquaintances unique to China has been gradually broken. The Chinese-style relationship networks influenced by socio-cultural and social policies and dominated by kinship priority acquaintance networks are influenced by the independence priority acquaintance networks in the context of modernization [45]. Currently, many residents consider the community as just a communal living area, and do not consider the community as a community of interests for all citizens; therefore, they are unable to form a sense of community identity, and have a low level of connection to the interests of the community, which causes them to be indifferent and less motivated to participate in community affairs [24]. At the same time, due to the needs of daily life and work, most residents in the community do not have the time and energy to ask questions about community affairs; consequently, social ties among residents, between residents and the community committee, and between residents and the community, are weak [46].

Second, the participation behavior is more utilitarian. On the one hand, ordinary residents do not spend much time and energy learning the knowledge needed to participate in community governance, and it is difficult to obtain or precisely understand information about the target of governance. The adequacy and effectiveness of public participation is also affected by the availability, accessibility, and ability to assess the skills and knowledge necessary to deal with complex public governance issues [47,48]. On the other hand, the returns to participation are low. Due to the lack of social capital support, communities

receive limited financial support, making some public participation only temporary voluntary participation and not sustainable in the long run [7]. Only when personal interests are severely compromised will individuals take the initiative to speak out or take active steps to solve the problem at hand, and once the matter is resolved, the status quo ante is restored.

Third, the participation process is mostly dominated by the Xiangxians. For large-scale public participation, there is a lack of institutional guarantees for fair participation [13]. In practice, public opinions are more often collected, screened, and integrated in an informal form, and then submitted to government departments in a non-standardized textual form. Therefore, there is a risk that the public's wishes are misinterpreted, ignored, or partially interpreted in the process of opinion collection and transmission. Meanwhile, due to the lack of standardization in the selection of public participation representatives, candidates are usually community activists, senior intellectuals, or association leaders, who poorly represent the needs of the entire community residents. The public opinion and opinion feedback mechanisms controlled by the Xiangxians creates the conditions for the "spiral of silence" phenomenon, and the chronic free-riding behavior greatly curbs the motivation and continuity of public participation.

Fourth, the sense of gain from participation is low. Public participation in community governance is mostly passive in terms of form and non-political in terms of content, which actually makes it difficult for the public to influence decision-making deliberations. The process of public participation in influencing the decision-making process consists of several links, including whether the public fully expresses their true opinions; how the opinions are transmitted to the decision-making department; how the decision-making department faces, integrates, and responds to the diverse opinions of the public; how reasonable public opinions are adopted or adjusted; and how the decision-making department provides feedback to the public on the resolution. However, under the current system and regulations, public participation is narrowly equated with the right to express opinions in the process of entering the decision-making process. As a result, there is a huge gap between the expectation of participation and the reality of participation, which makes the public doubt the value and role of participation and lack sufficient sense of efficacy.

*4.2. Social Organizations*

The participation of social organizations is an improvement in community governance, helping to promote the equalization of basic public services [49] and rebuild community attachment [50]. Although social organizations are not profit-oriented enterprises, they introduce business logic into community governance. Through questionnaire surveys, one-on-one interviews, and field visits to community residents, we have observed that the specialized capabilities of social organizations function in community governance in three main ways: (1) community needs mining, i.e., helping community committees identify residents' needs and helping residents provide feedback and persuade them to reach a consensus; (2) informal organization cultivation, i.e., using community affairs as ties, guiding youth groups and marginalized groups to integrate into community life [51], and promoting the formation of healthy neighborhood relationships in the community [52]; (3) linking and coordinating social capital, i.e., social organizations take the project as a guide to coordinate the capital investment and return of stakeholders, bridging the connection between community residents and other participating subjects [53].

However, there are certain functional limitations to the participation of social organizations. First, the distribution and level of specialization of community social organizations varies according to the level of geographical development. Since the scale of social organization service development depends on the financial capacity of local governments [54], social work in China is mainly concentrated in urban areas, and is neglected in rural areas; it is mainly concentrated in developed coastal areas, while community work in less developed inland areas lags behind [55]. This leads to the fact that the older neighborhoods lack social capital, making it more difficult to obtain quality external support.

Second, the participation of social organizations lacks stability. Compared with other participating entities in the community, social organizations are external participants. Generally speaking, the presence of social organizations enters with the development of the project and exits with the end of the project. Therefore, each social organization entering the community needs to re-establish a trusting relationship with the community residents; this mechanical repetition reduces the efficiency of social work, and is not conducive to cultivating deep and intimate partnerships.

Third, the social empowerment of social organizations is low. Due to historical factors, social work in China started late. Large-scale state intervention played a key role in the rapid diffusion of social organizations and social work [56], but this also objectively resulted in the consequence of poor autonomy of social organizations in China [57]. This has, to some extent, led to social organizations becoming subordinate to the administration.

*4.3. Community Committees*

In recent years, the deepening of community governance in China comes from the top-down promotion of the government on the one hand, and the bottom-up response of community residents on the other; nonetheless, the administrative orientation is still dominant at present. As the end of the vertical management of the Chinese government, community committees are responsible for many administrative tasks, and are also the organizers of the party networks in the community. In general, community committees provide services to community residents in three ways.

First, the community committee should establish a network of party members. A network composed of retired cadres and party members is the most powerful weapon of the community committee. This group is highly organized, dedicated, and responsive to the political and social mobilization of the CPC, so it is easy for them to rally under the call of the community committee and make efforts or even sacrifices for the interests of the party. In addition, the group of party members generally has the ability to participate in community governance. The selection of CPC members requires a strict process, and the candidates are usually outstanding people from a certain group with a high level of professional skills. Moreover, candidates need to have a certain level of popular support, both in terms of having volunteered to serve the public many times and having received general recognition of their work from the public. Lastly, and most importantly, the learning system of the CPC requires that party members receive collective education every month, and this continuous learning process allows the party members to shoulder the heavy responsibility of community governance.

Second, the community committee needs to gradually establish a volunteer team. The main challenges that affect community volunteers include lack of financial rewards, fragile relationships, vulnerability to burnout, tedious work, and lack of adequate community material support and spiritual appreciation for volunteering [58,59]. In some communities, a volunteer point system is used to attract residents to volunteer activities, and through cooperation with stores in the community, community volunteers can exchange their points for free or low-cost purchases in certain stores. Meanwhile, the community will issue commendation certificates to the volunteers and publicize the deeds of outstanding volunteers on the community publicity boards and regional newspapers. These incentives are the main initial reason to initiate and maintain volunteer activities [60].

Third, the community committee should provide direct resources. With community funding as the cornerstone, community committees can fund community beautification and renovation projects in the form of project outsourcing. At the same time, with the further development of the community development concept, the functions of community committees have undergone profound changes. For example, the community social enterprise being piloted in Chengdu is a major innovation to support diversified community participation. Community social enterprises are specific economic organizations that are wholly owned by community committees as specialized legal persons of grassroot mass

self-governance organizations, which carry out business management and the proceeds are used to continuously feed the community and promote community governance.

However, the mobilization target of community committees still has difficulty in covering the whole community, and their means of mobilization are more limited; worse still, a gap has appeared between community committees and residents. The value bias of public service provision. Influenced by national policies and government performance assessment, street offices, and community committees tend to provide community benefits to retired cadres, veterans, and disadvantaged groups, but for the vast majority of ordinary residents, these benefits and services have nothing to do with them, thus increasing their sense of alienation and non-belonging.

Furthermore, it has spawned the alienation of resident autonomy through the party's approach to social control and mobilization. In practical terms, capable community members are generally popular, educated, and capable potential social elites cultivated by the CCP. In the process of community governance, the discourse between elite residents (generally party members) and ordinary residents is not equal. The limited representation system, complicated voice channels, and administrative mechanisms for expressing opinions make most people tend to play the role of the silent in community governance. The high reliance on party members' network hinders the transformation of people's self-management and self-service consciousness. This "dominant" institutional and behavioral model is highly effective in mobilizing people in public safety emergencies, but it is not conducive to realizing residents' self-governance in daily public affairs governance.

### 4.4. Enterprises and Commercial Establishments in the Community

Under the trend of the "turn to community" movement [61], enterprises and commercial establishments This refers to profitable establishments that belong to the same community or street jurisdiction, such as stores, restaurants, hotels, office buildings, factories, etc.) are gradually becoming more involved in community affairs from being invisible "bystanders". However, corporate participation in community governance is very limited to routine affairs such as environmental transformation, convenience services and joint defense. Corporate participation is generally passive and mostly used as a resource provider [62], making it difficult to build a sense of identity with the community. As a potential participant in community governance, there are also natural barriers to corporate participation [63].

First, there is a conflict of interest between commercial establishments and residents in the community. Traditionally, commercial establishments in the community usually interact with residents through fundraising, providing space or convenience services under the initiative of the community committee [64], maintaining a fragile and detached relationship. With the overlap between the business space and the residential space, commercial establishments and residents in the community are often in an adversarial or even hostile relationship due to pollution problems such as noise pollution, light pollution, smoke, and wastewater, as well as public problems such as tight parking spaces, complex staff structure, and safety hazards caused by too many mobile people.

Second, enterprises in the community are always "outsiders". Although the business activities of commercial establishments are within the community, not all the employees of commercial establishments live in the community. As a result, community affairs are even more irrelevant to them. They have neither the incentive to manage community affairs, nor the empowerment to interfere in community affairs. The participation of commercial establishments is often the result of a multi-stakeholder game, and the leading role is usually played by the community committee.

Third, the profit-seeking nature of enterprises leads them to pursue the maximization of their own interests in the process of community governance [65]. Corporate participation usually comes with certain commercial "conditions", such as advertising for sponsors, using their products, and access to community publicity, which actually "force" the government to endorse them. Purely commercial activities that overly pursue commercial value can cause

the public to question the nature of the event. Under the government's "nanny" model of governance, corporate engagement lacks innovative models that combine community culture, corporate social responsibility, and corporate business interests [66].

## 5. Conclusions and Discussion

The practice of the community proposal model in Chengdu is a microcosm of contemporary Chinese urban community governance innovation. Community proposals integrate mobilized participation and autonomous participation, organizational participation, and unorganized participation, which greatly stimulate the public's willingness to participate in community governance, broaden the channels for public opinion expression and feedback, enhance the public's initiative and sense of efficacy, and raise the level of public awareness and ability in democratic consultation. However, this community proposal mechanism is a local experiment opened under the reform drive of the China Experimental Zone for Community Governance and Service Innovation. Under the policy guarantee and financial support of the government, as well as the pressure of performance assessment, the community proposal is the result of a policy-oriented approach. It certainly has many benefits, but it is not known whether it will continue to deepen and carry out, whether it will be recognized by higher governments or other local governments, and whether it can be promoted on a large scale.

The community proposal model with the participation of diversified subjects is still dominated by community committees. When Xiangxians, community organizations, and enterprises participate in community proposals, they need to cooperate with the community committee actively or passively. Therefore, public participation is active in the "minority". The balance of power between the government and social actors in the process of community governance is still a difficult issue [67]. Under a pressure-based system, community governance reforms are often passed downward and cascaded through administrative tasks. When administrative power and social capital are abundant, a well-designed governance model can achieve synergy among multiple participating actors, such as government policy guidance and neighborhood collaboration in the context of crisis [68]. On the contrary, in most spaces and fields, the top-down governance pressure will make the grassroots fall into a state of passive obedience and rigid implementation, while limiting the creative ability of unofficial participating actors [69].

At the same time, it should also be observed that de-administration is still a key factor in stimulating public participation. The involvement of local governments and party organizations in grassroots community governance helps prevent the problem of selective implementation in the context of street bureaucracy or ambiguous policies, but it also further strengthens the negative impact of the hierarchical structure on policy implementation. The result that policy resources and social capital for community governance innovation may be influenced by "political potency" to flow to the model areas, thus exacerbating the differences in resource endowment between districts. Once policy resources are withdrawn, it is difficult for policy-dependent demonstration areas to develop endogenous governance participation structures. So, how should local governments actually decentralize in order to ensure the proper implementation of community governance policy innovations in terms of procedure and order? The insights from the Chengdu community proposal are: first, to strengthen the value orientation of public participation and weaken the participation of administrative agencies at the level of "technical governance"; second, to take formal organizations as the entry point and cultivate the participation capacity of informal organizations, so as to catalyze public understanding and action awareness of community governance; third, the government should "certify the procedures" of the gradually formed participation of multiple subjects, so as to construct a participation mechanism in line with community characteristics, public interests and democracy.

Furthermore, how can multiple subjects exercise their participation rights in a more standardized, legalized, and scientific manner? In the absence of a strong leadership role, the governance participation of multiple subjects may deviate from the original purpose

of collective action and instead fall into a more conflicting, disorderly, and inefficient governance state. In this case, the increase in governance power does not change the low-quality participation behavior. Therefore, the first problem that needs to be solved for the governance participation of multiple subjects is the formation of reasonable appeals and the gradual expansion of the cooperation space of multiple participating subjects, such as community residents, government agencies and enterprises, based on the predictable economic interests and potential multiple values.

The practice of community proposals has given new insights into the urban community governance in contemporary China. Community governance, with the participation of diversified subjects, can effectively activate various types of community capital, which is the direction of transformation of China's urban community governance [70]. The optimal path of community governance innovation for diverse subjects in China lies in tapping community social capital, expanding the dimensions of social capital, and creating a sustainable social capital transformation mechanism. Good community governance is a process of creating a system of rules, which contains both institutional and emotional resource guarantees. Institutional resource guarantees refer to the construction of institutional social capital that allows individual actions to be shifted to collective and orderly cooperation, strengthens the stability and reliability of public participation in governance, and creates a social environment for building a sustainable performance growth path. In contrast, emotional social capital can contribute to the formation of organizations, maintain ties among members, and continuously create flexible cultural values to provide environmental support for community governance [71]. This study argues that the mechanism of community proposals can realize the creation of emotional social capital and institutional social capital, providing many possibilities for public participation in community governance. This is a new attempt to reshape the "structure–process" mechanism of participation of multiple subjects under the socialist system with Chinese characteristics.

**Author Contributions:** Conceptualization, D.W. and S.L.; methodology, S.L.; software, S.L.; validation, D.W. and S.L.; formal analysis, D.W.; investigation, D.W. and S.L.; resources, D.W.; data curation, S.L.; writing—original draft preparation, S.L.; writing—review and editing, D.W. and S.L.; visualization, S.L.; supervision, D.W.; project administration, D.W.; funding acquisition, S.L. All authors have read and agreed to the published version of the manuscript.

**Funding:** This research was funded by [National Social Science Foundation of China] grant number [22CDJ042].

**Conflicts of Interest:** The authors declare no conflict of interest.

## Appendix A

**Table A1.** The scoring rules for community proposals.

| Evaluation Category | Evaluation Content | Rules for Evaluation | Points |
|---|---|---|---|
| Basic points (40 Points) | Case topic | The topic is novel and close to the theme. | 4 |
| | Text framework | The framework is complete and logical. | 5 |
| | Content elaboration | Explain the time, place, people, measures, results, and focus on the measures and results. | 12 |
| | Proposer | Reflect the proposer, the proposer is divided into organizations and individuals. | 4 |
| | Proposition issues | The submitted topics are reasonable and belong to the public issues, including construction, renovation, public security, community service, autonomy, and autonomous co-governance. | 5 |
| | Submission method | Submit using the online applet for community proposals. (Just submit a screenshot). | 5 |
| | The role played by the proposal organization | Submit to the four-level community proposal organization, and accept and promote the solution. | 5 |

**Table A1.** *Cont.*

| Evaluation Category | Evaluation Content | Rules for Evaluation | Points |
|---|---|---|---|
| Advanced points (44 Points) | The role of party building | Give full play to the role of Party organizations at all levels in the proposal work, leading, and give play to the vanguard and exemplary role of party members. | 8 |
| | Diversified proposal subjects | There are multiple participants in solving the proposal process, including party members, two representatives and one committee member (Representative of the Communist Party of China, deputy of the National People's Congress, member of the Chinese People's Political Consultative Conference), professional social workers, community workers, businesses, students, social organizations, self-organizations, schools, etc. | 8 |
| | Standardization of the proposal process | The community proposal process is standardized and complete, including proposing, accepting, discussing how to solve, implementing, feedback, etc. | 4 |
| | Consultation in the proposal process | The proposal is solved by consultation, including a consultation system, consultation form, consultation rules, and complete consultation process. | 8 |
| | Effectiveness of the proposal | Effectiveness is evident, and there is an understanding of the proposal mechanism. Community proposals at all levels have evaluations from higher-level community proposal organizations. | 16 |
| Characteristic points (16 Points) | Innovations in the proposal | The content of the proposal has characteristics such as remediation, public safety, etc.; the subject of the proposal and the subjects involved in the proposal have characteristics, such as students, merchants, units in the district, etc. | 8 |
| | Scalable experience in the proposal | Specific solution practices are refined and can form replicable experiences and models for other streets, communities, and neighborhoods to use. | 8 |

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
