# Peer review of "Innovation of Contemporary Chinese Urban Community Governance under the Perspective of Social Capital: Participation of Multiple Subjects Based on Community Proposals"

_sustainability, doi:10.3390/su15010093_

Round 1

Reviewer 1 Report

This is a very high-quality research on the community governance models in China from the social capital perspective. I have two major questions related to the research methodology:

1.     Page 7 -  in the methodology section, how the researchers examined such extensive data? Please give more details about the data analysis methods and tools.

2.     It is difficult to understand how the authors select the example of each community (community A, community B, etc.) to exemplify and describe the features of four community governance models among 30 cases. It would be good to explain the selection strategy. Also, how can the four examples be generalizable when looking at the 30 cases?

3.     A comment on the introduction of some incomplete sentence, statement or sub-headings within the text, such as on page 2, the lines between 65-97.

-        (E.g. Line 70) First, the study of participating subjects and their functions. – this is neither a sub-heading nor a sentence.

Authors use this way of introducing the sub-topics in different parts of the manuscript. I would suggest the authors revise the such text in the manuscript either as a sub-heading (in italic form) or a sentence within the text. 

Author Response

We really appreciate you for your carefulness and conscientiousness. Your suggestions are really valuable and helpful for revising and improving our paper. According to your suggestions, we have made the following revisions on this manuscript:

  1. In the second part of the manuscript, we added a description of the specific methods of data analysis and the analysis tools used. Please see Page 12, 13.
  2. In 2.2, we have added the basis of case selection, elaborated the representativeness of the four cases, and added the Scoring Rules for Community Proposals in Appendix 1. Please see Page 12 and 43.
  3. Figure 2 is added to assist understanding. Please see Page 7 and 8.
  4. The relevant incorrect statement in the whole article has been changed to the completed sentence.

All the modifications are marked in red.

Reviewer 2 Report

Most of the references cited are more than 5 years. Please add more recent references. The conclusion section is too elaborative.

Please highlight the key findings of the study. 

Abstract lacks an explanation on the methodology adopted in the study. 

Author Response

Thank you for reviewing our manuscript and offering valuable advice. In accordance with your suggestions, we have made the following revisions to our manuscript:

  1. Added the latest literature from the last five years.
  2. In Part V, unnecessary content has been eliminated and the conclusions of the study have been concisely stated. Please see Page 34 to 37.
  3. Added an explanation of the research methodology of this manuscript in the abstract.

All the modifications are marked in red.

Reviewer 3 Report

The paper aims to discuss evolutions and prospects in regards to the Chinese urban governance models, under the perspective of social capital and based on community proposals. The structure of the manuscript is clear and the included figure and table are considered necessary for the reader to follow the research framework and the comparative analysis of the proposed four models. The topic itself is interesting, however I suggest several revisions before having it considered for publication.

o  First of all, the authors make use of different terms in relation to different(?) aspects of governance, which are not made clear. Specifically, according to my understanding the terms ‘social governance’ and ‘community governance’ are used without the actual differences between them being clear. At the same time, the use of several similar notions throughout the text such as ‘pluralistic governance’, ‘grassroots governance’, ‘collective governance’, ‘joint governance’ etc. is not followed by the proper understanding of the content / meaning and the reason for using all these different notions. Especially the term “government governance” is not familiar to the wider audience (including myself) and would need to be explained. Based on the previous remarks, I believe that the authors should revise the whole manuscript and remain loyal to the use of the appropriate terms / notions.

o  Similar is my concern on the use of the terms “community renewal” and “community revitalization” in the first paragraph of the introduction.

o  According to my understanding, the structure process analysis framework is not clear. For example, it is not clear what is new about “emotional social capital” in comparison to “emotional capital”, therefore the authors should expand on it. Also, it is not clear what the “network of emotional support” (lines 205-206) is about. Finally, in Figure 1 I suppose “participation capital” is the right term instead of “participating capital”.

o  The China’s community-based system referred to in lines 258-259 is not necessarily known to the readers, so I suggest that the authors offer some descriptions / explanation. I also suggest to provide a more clear structure of the process described in lines 267-273 (maybe a figure would be of help). 

o  I am not sure whether Table 1 has to precede the analysis of the models rather than follow it. Also, I cannot understand why some models are referred to as “models” and others as “modes”.

o  Regarding model 1, it is not made clear from its description why it is an “elite dominated style”. The authors should also elaborate on that.

o  A paragraph should be included describing the process of leading to these four models based on the cases examined, as well as give some data on the frequency of the models, i.e. how many research cases fall under each of the four of them.

o  The manuscript needs to go through professional proofreading, in order to correct several (mainly) syntax issues and improve the coherence and understanding of its content. For example (and INDICATIVELY) there are:

-       - Syntax issues (e.g. lines 281-285 (reviews cannot “be” either representatives or experts), 307-308, 342-344, 652-653, 1050-1051 (institutional resources cannot “be” construction of capital), 1089)

-       - Very large sentences, making understanding of the actual meaning / content not easy (e.g. lines 180-188, 217-223, 443-454, 668-675, 1018-1024)

-        - Typos (e.g. line 568)

-      -  Unnecessary repetitions (e.g. 603-605, 725)

o  The conclusions section ends up discussing flexibility and adaptability, however such conclusions do not derive from the research conducted and described previously in the manuscript. Also, it is not clear what “effective withdrawal” means and of what (the same also stands for the “effective exit” referred to in the abstract).

o  Finally, some additional remarks:

-       - Line 89: The authors should mention any limitations that they observe.

-     - Lines 167-189: The authors refer to one problem and one issue, but these are rather questions.

-      -  Line 188: What is the “dilemma” here?

-       - Line 208: It is not clear what “closing process” is.

-      -  Line 1116: What is the “dilemma” here?

-       - Line 1120: What is interest rationality?

Author Response

We really appreciate you for your carefulness and conscientiousness. Your suggestions are really valuable and helpful for revising and improving our paper. According to your suggestions, we have made the following revisions on this manuscript:

  1. In view of a series of problems pointed out, such as improper use of words, inconsistent use of terms, etc., we have made modifications, unified the use of words, and added notes to some terms. Including but not limited to “social governance”, “community governance”, “pluralistic governance”, “grassroots governance”, “collective governance”, “joint governance”, “government governance”, “model”, “participating capital”, etc.

  1. We have explained the China’s community-based system referred to in lines 258-259. And we have added an introduction to the community proposal process in lines 267-273. Please see Page 9, 10, 11 and Appendix 1 on Page 43.

  1. The position of Table 1 has been adjusted to the end of Part III.

  1. The concept of elite governance was replaced by a more appropriate concept of Xiangxian governance, and an explanation was given. Please see Page 13.

  1. The structure-process analysis framework has been modified. And the discussion on individual social capital, collective social capital, emotional social capital and institutional social capital has been added, and Figure 1 has been modified. Please see Page 7 and 8.

  1. The description of case selection and its representativeness is added, and Figure 2 is added to assist understanding. Please see Page 7 and 8.

  1. The conclusion is rewritten, and irrelevant content is deleted. Please see Page 34 to 37.

  1. We have revised the unknown semantics. Including but not limited to “limitations” in Line 89, “problems” in Lines 167-189, “dilemma” in Line 188 and 1116, “closing process” in Line 208, etc.

All the modifications are marked in red.

Round 2

Reviewer 3 Report

Dear authors, the manuscript has been properly revised.

I detected some typos (e.g. line 765, where 'community' needs to start with capital letter) but I suppose these will be corrected during the proofreading process of the journal.